# Adverse Events of Mind-Body Interventions in Children: A Systematic Review

**DOI:** 10.3390/children8050358

**Published:** 2021-04-29

**Authors:** Meagan Lyszczyk, Mohammad Karkhaneh, Kerri Kaiser Gladwin, Martha Funabashi, Liliane Zorzela, Sunita Vohra

**Affiliations:** 1Department of Medicine, University of Alberta, Edmonton, AB T6G 2G3, Canada; mlyszczy@ualberta.ca; 2Institute of Health Economics, Edmonton, AB T5J 3N4, Canada; mk4@ualberta.ca; 3Department of Pediatrics, University of Alberta, Edmonton, AB T6G 1C9, Canada; kerrik@ualberta.ca (K.K.G.); funabash@ualberta.ca (M.F.); zorzela@ualberta.ca (L.Z.); 4Division of Research, Canadian Memorial Chiropractic College, Toronto, ON M2H 3J1, Canada; 5Department of Psychiatry, University of Alberta, Edmonton, AB T6G 2B7, Canada

**Keywords:** mind-body interventions, children, safety, adverse events

## Abstract

Mind-body interventions (MBIs) are one of the top ten complementary approaches utilized in pediatrics, but there is limited knowledge on associated adverse events (AE). The objective of this review was to systematically review AEs reported in association with MBIs in children. In this systematic review the electronic databases MEDLINE, Embase, CINAHL, CDSR, and CCRCT were searched from inception to August 2018. We included primary studies on participants ≤ 21 years of age that used an MBI. Experimental studies were assessed for whether AEs were reported on or not, and all other study designs were included only if they reported an AE. A total of 441 were included as primary pediatric MBI studies. Of these, 377 (85.5%) did not explicitly report the presence/absence of AEs or a safety assessment. There were 64 included studies: 43 experimental studies reported that no AE occurred, and 21 studies reported AEs. There were 37 AEs found, of which the most serious were grade 3. Most of the studies reporting AEs did not report on severity (81.0%) or duration of AEs (52.4%). MBIs are popularly used in children; however associated harms are often not reported and lack important information for meaningful assessment.

## 1. Introduction

Mind-body (MB) interventions are types of complementary therapies designed “with the intent to use the mind to affect physical functioning and promote health” [1]. They are a diverse group of modalities including biofeedback, hypnosis, and meditation, and have been utilized at least once by 5.3% of children in the United States aged 4–17 [2]. The popularity of MB interventions in pediatrics is increasing [3,4,5], as evidenced by a repeated survey (2007 and 2012) that demonstrated that their use had increased from 2.5% to 3.2% amongst children 4–17 years old [6]. Hypnosis and biofeedback are amongst the most commonly utilized MB therapies and are used to treat a variety of conditions including chronic pain, headache, enuresis and IBS [2,3]. Advantages of these therapies include their non-invasive nature, cost-effectiveness, and promotion of self-efficacy in pediatric patients that can contribute to improved coping skills and resiliency [7].

Safety of any health intervention is of great importance for patients and clinicians [8]. Safety can be assessed through monitoring for adverse events (AE), which are defined as “any unfavorable and unintended sign, symptom, or disease temporally associated with the use of a medical treatment or procedure” [9].

There is a recognized need for improvement in the assessment and reporting of harms [10,11]. Clinical trials are designed to collect, evaluate, and report harms associated with interventions [12]. However, randomized controlled trials (RCT) cannot be relied on to capture many AEs associated with an intervention due to inadequate sample sizes and trial duration to assess long-term harms, homogeneous populations that do not represent intervention use in real-world practice, and lack of harm assessments as primary objectives [8,13,14,15]. Unpublished supplemental data from RCTs and published data from controlled observational studies including case-control and cohort studies should additionally be scrutinized for AEs [8]. Uncontrolled studies including case reports and case series can also identify AEs, but are limited by a high probability of selection bias and lack of direct causal association between AEs and the intervention [8].

Systematic reviews of the literature seek to provide a high quality, unbiased, and comprehensive summary of evidence [16], yet less than 10% report harms as a primary objective [17]. Reviews of this kind synthesize valuable data regarding AEs, which allows health care practitioners and patients to make informed decisions with consideration to an intervention’s harms and benefits [17].

While the interest in and use of MB approaches is increasing, there are limited formal data synthesized about their potential harms. The primary objective of this review was to systematically identify and synthesize available data on the adverse events associated with MB therapies in pediatric patients.

## 2. Materials and Methods

The PRISMA guidelines were followed to develop and conduct this systematic review [18].

### 2.1. Data Sources

A comprehensive search strategy was developed in conjunction with a health research librarian and run in five electronic databases. The following databases were searched from inception to August 2018: MEDLINE (1946–2018), EMBASE (1974–2018), CINAHL (1937–2018), the Cochrane Database of Systematic Reviews (2005–2018), and the Cochrane Central Registry of Controlled Trials (CENTRAL) (1991–2018). Additional references were obtained by hand-searching the Google Scholar web search engine. A copy of the Medline search strategy can be found in Appendix A.

### 2.2. Study Selection

After removing duplicates, two review authors (ML and KKG/MF) screened the titles and abstracts of identified citations. Full text articles deemed to be potentially relevant were retrieved for full review and assessed by two independent review authors (ML, KKG/MF) using a predetermined set of inclusion criteria: (i) primary investigation/report (i.e., not a review, commentary); (ii) pediatric participants (0 to 21 years of age); and (iii) studied a MB intervention (see Appendix B). Interventional and observational studies including RCTs, CCTs, single-arm experimental, prospective cohort, case-control, and controlled before and after studies were included and evaluated for any assessment of safety/AE. These studies were categorized if they (i) assessed safety and reported AE, (ii) assessed safety and reported no AE, or (iii) did not report on safety or AEs of the intervention. Case reports, case series and any remaining observational studies were only included if they reported an AE. Non-English articles were excluded. Disagreement was resolved by discussion and if required, consultation with a senior review author, until consensus was reached.

### 2.3. Data Extraction

Data was extracted by a review author (ML) using a structured data-extraction form and verified by second reviewers (CB, MF). General (study methods, settings, age, sex, etc.) and specific (AEs, timing, etc.) information was extracted from the included studies. If further information was required, the corresponding author of the study was contacted. Disagreement was resolved by consensus.

### 2.4. Data Synthesis

The severity of the AEs was assessed by two reviewers (ML and MF) using the Common Terminology Criteria for Adverse Events (CTCAE) [9]; discrepancies were resolved by a third review author. The following categories were used: grade 1 (asymptomatic/mild symptoms, intervention not indicated), grade 2 (moderate, limit age-appropriate activities or local/noninvasive intervention required), grade 3 (severe, hospitalization indicated, but not immediately life-threatening), grade 4 (life-threatening) and grade 5 (resulted in death). Since this study did not investigate the effectiveness of interventions, neither risk of bias was evaluated nor was a meta-analysis performed.

## 3. Results

After screening the titles and abstracts of 13,048 citations, 1455 full text articles were retrieved, of which 1014 were excluded and 441 were included as primary pediatric MB intervention studies (Figure 1).

Of these 441 studies identified, 254 (58%) were experimental and 187 (42%) were observational studies.

Within the 441 studies, only 64 (14.5%) explicitly reported presence/absence of AEs or assessed safety, of which 43 (67.2%) reported that no AE occurred, and 21 (32.8%) reported AE(s) (see Figure 1, PRISMA flow chart for details). The most common types of MB intervention studied were biofeedback and hypnosis, with 180 and 82 studies respectively (Appendix B and Appendix C).

Of the 254 experimental studies (*n* = 7213), 200 (*n* = 5647) did not report on AEs (if occurred/did not occur or if they were assessed). Of the 54 (*n* = 1566) studies that reported on AE, 43 (*n* = 1405) reported that no AEs were found and 11 reported AEs following MB interventions. Most of the studies reporting adverse events did not report on severity (81.0%) or duration of adverse events (52.4%).

### 3.1. Adverse Events

Of 21 studies reporting an AE, 11 (*n* = 208) were experimental and 10 (*n* = 406) were observational (Table 1). These studies reported one to four AEs each, for a total of 37 AEs (Table 2).

### 3.2. Adverse Events of Pediatric Mind-Body Interventions by Severity

Using CTCAE criteria for rating severity of AEs [9], three were rated as Grade 3, three as Grade 2, and 20 as Grade 1 (Table 2). There were no Grade 4 or 5 AE amongst reported MB AEs. We were unable to evaluate the severity of the remaining 11 AEs due to insufficient information provided in the article.

#### 3.2.1. Grade 3

The most serious AEs identified were Grade 3, reported in three patients. One event was a case of unresponsiveness to verbal communication in a 13-year-old female with acute lymphoblastic leukemia who had initially been hospitalized with probable toxicity to her chemotherapy. She had utilized self-hypnosis in hospital for symptom control and could not come out of her hypnotic state independently. This necessitated transfer to an acute care unit for closer observation as her hypnotic state was misinterpreted as a possible neurological deterioration. The therapist who had taught her self-hypnosis facilitated her return to an alert state [26]. The second Grade 3 AE was an intra-abdominal bleed in a hemophiliac 18-year-old male who was utilizing hypnosis as a means of reducing bleeding and pain. Several hours prior to the development of the bleed, he had recalled two prior bleeds at the same site during a session of hypnosis. Treatment was provided by his hematologist; no specific surgical intervention was required. The study authors postulated the bleed may have been related to an ability of hypnosis to affect vasculature and blood flow (Table 2) [24]. The last grade 3 AE was a tibial fracture sustained by a 14-year-old female while assuming a yoga position in a school physical education class. The fracture was reduced and required a cast but did not result in a hospital admission [33].

#### 3.2.2. Grade 2

There were three AEs rated as Grade 2. One of these AEs was the onset of mesial temporal lobe epilepsy in an 18-year-old female with no known risk factors after lifelong transcendental meditation practice. She had neurological assessment (MR and EEG) but hospitalization was not reported. The authors cautioned that there is insufficient evidence to definitively establish or disprove that meditation may precipitate seizures [29]. Another Grade 2 AE involved an 18-year-old female who had an apparent epileptic seizure while practicing hypnosis. A subsequent EEG was normal, and it was thought to likely be a spontaneous event given the absence of a personal or family history of seizures [25]. The last Grade 2 AE identified was a 17-year-old male who had increased symptoms of depression and reduced therapeutic engagement after biofeedback and progressive muscle relaxation for tension headaches (Table 2) [36].

#### 3.2.3. Grade 1

There were 20 AEs rated as Grade 1 (mild) (Table 2). Seven adverse events were associated with the practice of hypnosis, including blue-tinted vision with a concurrent penile erection [21], increased anxiety, dissociated states, depersonalization phenomena [22], physical discomfort [23,27], and retroactive amnesia [25]. Relaxation had five adverse events associated with it, including four instances of increased betamimetic medication use [38] and an increase in tic frequency [31]. There were also four adverse events related to biofeedback: three cases of intervention-induced anxiety [19,20], and one case of foot pain [19]. The remaining four events associated with yoga, imagery, and a multi-modal MB intervention, were: dizziness [35], emotional distress and physical shaking [28], and chest pain [37], respectively.

### 3.3. Unclear Severity

Eleven AEs could not be rated for severity due to insufficient information (Table 2) [30,32,34,39].

## 4. Discussion

To the best of our knowledge, this is the first systematic review examining the safety of all pediatric MB interventions. Of potential concern, the vast majority of primary pediatric MB studies (85.5%) did not report if/how safety was measured. It is important to distinguish the absence of occurrence of AEs from the lack of their reporting. These are not equivalent, and lack of reporting can create bias during the assessment of an intervention if only its efficacy, or benefits, are evaluated and reported [10,11].

While there are systematic reviews that have extracted AE information on individual MB therapies [40,41,42,43,44,45,46,47,48,49,50,51], few of these have addressed AE as a primary objective [52,53,54,55]. Within the reviews with AE as a primary outcome, only three adverse events related to MB therapies were captured [34], in comparison to our review which was able to identify 37 adverse events. This synthesis helps to fill the existing gap in pediatric MB therapy research.

The majority of AEs identified were minor in nature; however, many of the studies did not provide pertinent details such as event duration or patient outcome. Incomplete reporting is significant as it hampers the ability to assess causation between an intervention and AE [56]. Additionally, poor reporting at the primary study level impairs the ability of systematic reviews to provide a balanced assessment of an intervention’s efficacy and harms. Regulatory frameworks to monitor the practice of complementary therapies would be beneficial [57], as at present there are no established standardized methods for assessing harms associated with MB interventions [58].

The absence of more serious events (Grade 4 and 5) is encouraging, but our ability to accurately estimate adverse events associated with these interventions is limited. While RCTs are regarded as the gold standard of research to assess efficacy, they report harms poorly [10,11,17], and are often statistically underpowered to detect rare, serious events [59]. This review is a first step in synthesizing best available information, to better plan future prospective research to identify and report AEs associated with pediatric MB therapy use.

While the majority of adverse events were reported in teenagers, there are insufficient data to make conclusions about AE profiles for different ages of children. Future studies should consider exploring age differences in adverse events associated with MB interventions.

MB interventions are popular and there is growing evidence for clinicians to support their use in children and youth to reduce stress, anxiety, and depression [60,61,62,63]. We recognize all health interventions have the potential for benefit and for harm. If patients experience an adverse event, there is value in reporting this. Our goal is to promote an evidence-based approach when considering health interventions, including weighing potential benefits and harms of various treatment approaches to determine which is preferred for an individual.

Assessing causality was limited as many of the identified studies were uncontrolled studies. While these are useful for evaluating adverse events related to an intervention, they are limited by a high probability of selection bias and therefore cannot confirm causation between an intervention and associated AEs [8]. Controlled trials, while the best design to examine causation, are hampered by an inability to detect rare, serious events [64]. Incomplete adverse event reporting further hampers the ability to assess causation between an intervention and AEs [56].

An important limitation of this review is the lack of adverse event reporting in the included studies, which limits the full understanding of the safety of pediatric MB interventions. Lack of adverse event reporting is not equivalent to lack of occurrence—lack of reporting could mean that: (i) no adverse events occurred; (ii) adverse events were not sought/assessed; or (iii) adverse events were identified, but not reported. Systematic reviews are only as reliable as the data presented in the included studies.

One potential limitation of this study is focusing on only English-language articles. Reviewing studies written in additional languages may provide more information and decrease the chance of selection bias [65] but was not feasible. Additionally, we were unable to obtain an estimate of adverse event rates for MB interventions, due to the lack of denominator data.

This study has multiple strengths, including that to our knowledge this is the first systematic review to summarize reported AEs associated with MB interventions.

Additionally, all study types from case reports to RCTs were included. Data regarding the reported AEs was further enriched by rating their severity with standard criteria. Selection and information bias were further reduced by having two reviewers independently apply inclusion criteria to the retrieved full text articles and perform data extraction/verification. Adverse events data can also be affected by publication bias, as less attention has been given to adverse events in comparison to efficacy of interventions [8,16,17,66,67,68,69].

Mind-body therapies are popular and would benefit from improved reporting of associated adverse events. Like other fields [70,71], MB would benefit from the development and validation of tools to measure associated AEs. Active surveillance is another means of improving the identification and reporting of adverse events [72,73]. Only if AEs are known, can risks be mitigated and safety enhanced.

This review identified adverse events associated with MB interventions, the majority of which were mild. The lack of adverse event reporting in the majority of included studies warrants caution in interpreting these results, as lack of reporting does not necessarily mean lack of events. Observational research is the foundation for advancing patient safety and several scales exist to help assess the likelihood that an AE is attributable to an intervention [72,74].

As uncontrolled retrospective studies are vulnerable to bias [75], an emphasis should be placed on prospectively assessing MB AEs in controlled research, such that associations between interventions and AEs can be better understood.

## 5. Conclusions

MB interventions are commonly used by children, and while some mild (Grade 1) to moderate (Grade 2–3) adverse events have been reported, serious (Grade 4–5) AEs were not identified. One cannot assume lack of AE reporting is equivalent to lack of harm. There is a need for researchers and health care providers to assess and report adverse effects associated with pediatric mind-body therapies. Better quality information will help promote informed decision-making by patients and health care providers.

## Figures and Tables

**Figure 1 children-08-00358-f001:**
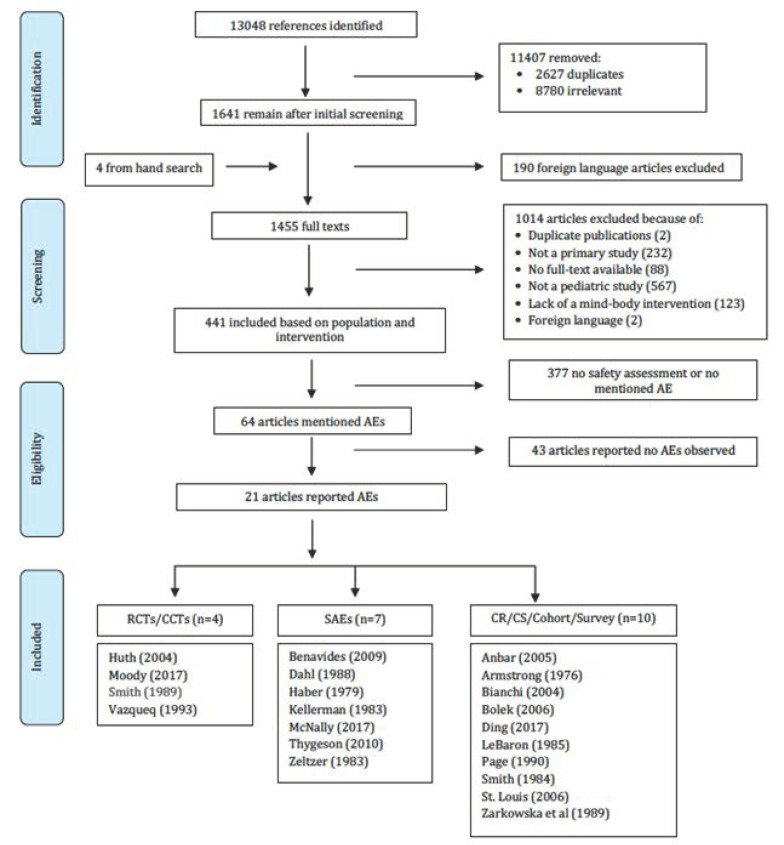
PRISMA diagram of study selection for pediatric mind-body adverse events. AE: adverse event; CCT: controlled clinical trial; CR: case report; CS: case series; RCT: randomized controlled trial; SAE: single-arm experimental study.

**Table 1 children-08-00358-t001:** Characteristics of the studies reporting mind-body adverse events in pediatrics.

MB Intervention	First Author, Year, Country	Study Design	# of Participants	Age Mean (SD), Range	Sex (% Male)	Reason for Seeking Treatment	MB Provider	Frequency and Length of MB Therapy	Limitation(s)
**Biofeedback**									
	Bolek (2006), USA [19]	Retrospective cohort	16	8.31 (5.15), 4–18	NR	To improve motor control (e.g., standing, sitting, head control, etc.)	Therapist	Patient-specific planning	No concurrent control group
	Dahl (1988), USA [20]	SAE	3	14 (1.7), 12–15	66%	To treat frequent refractory epileptic seizures	Psychologist	Patient-specific treatment	No concurrent control group, poor representation of the population
**Hypnosis**									
	Anbar (2005), UK [21]	Case report	1	15	100%	To improve adherence to cystic fibrosis therapy	Physician	On a daily basis for 7 years	Individual anecdotal report of AE
	Haber (1979), USA [22]	SAE	8	15.44 (1.55), 13–17	50%	To treat resistant obesity (to decrease food consumption)	NR	NR	No concurrent control group, poor representation of the population
	Kellerman (1983), US [23]	SAE	16	14.0 (1.6)	44%	To ameliorate discomfort and anxiety in adolescents with cancer	Pediatricians and psychologists	Training session and during procedure	No concurrent control group
	LeBaron (1985), US [24]	Case report	1	18	100%	To reduce pain, codeine usage, and bleeding associated with hemophilia	NR	5 months	Individual report of AE
	Page (1990), US [25]	Case series	2	18 (0)	50%	Nonclinical study volunteers	NR	NR	Individual reports of AE
	Smith (1984), US [26]	Case report	1	13	0%	To reduce procedural anxiety, muscle contraction, and headaches	Therapist	Utilized twice daily 4 days prior to hospitalization	Individual report of AE
	Zeltzer (1983), US [27]	SAE	9	14.2 (3.3), 10–20	58%	To reduce chemotherapy side effects (e.g., vomiting) in cancer patients	Psychologist	1–3 sessions prior to and during chemotherapy	No concurrent control group, different level of acceptance of hypnosis amongst participants
**Imagery**									
	Huth (2004), Netherlands [28]	RCT	36 (treatment)	9.42 (1.74), 6–12	44%	To reduce pain in tonsillectomy/adenoidectomy	Investigator	2–22 days prior to surgery and post operatively	Potential for children to over-report to please investigator, inability to provide sham treatment, inability to control pre-test pain equivalency
**Meditation**									
	St Louis (2006), UK [29]	Case report	1	18	0%	Practicing transcendental meditation since childhood	NR	Not clear but practicing since childhood	Individual report of AE
**Relaxation**									
	McNally (2018), USA [30]	SAE	26 (completers)	15.9 (2)	32%	To treat persistent post-concussive symptoms	Psychologist	2–5 sessions (45–60 min duration each)	No concurrent control group, findings may not be generalizable to other clinical concussion populations
	Zarkowska (1989), UK [31]	Case report	1	13	0%	To treat Tourette Syndrome in a cognitively delayed child	NR	Individual-specific schedule	Individual report of AE
**Yoga**									
	Benavides (2009), UK [32]	SAE	14	11.7 (1.5), 8.8–14.7	21%	Weight management and to improve self-concept/psychiatric symptoms	Yoga instructor	3 days/week for 12 weeks, 75 min sessions	Small sample size, lack of control, unable to fully evaluate long-term outcomes
	Bianchi (2004), Italy [33]	Case report	1	14	0%	Yoga in physical education class	Therapist	Once	Individual report of AE
	Moody (2017), USA [34]	RCT	35 (treatment)	NR, 6–20	40%	Sickle cell disease vaso-occlusive crises	Yoga instructor	Daily 30 min sessions, average 2.5 (1.6) sessions total	Randomization not blinded, small sample size, limited number of yoga sessions, single institution
	Thygeson (2010), US [35]	SAE	16	8.5 (1.75), 7–12; 15.4 (1.82), 13–18	63%	To reduce distress associated with diagnoses on hematology/oncology unit	Registered yoga teacher	Single yoga session	Recruitment issues (selection bias) due to lack of yoga experience among participants and parents
**Multiple**									
Biofeedback and progre-ssive muscle relaxation	Armstrong (1976), USA [36]	Case report	1	17	100%	Tension headaches	Therapist	NR	Individual report of AE
Biofeedback and Relaxation/Imagery	Smith (1989), US [37]	RCT	20 (treatment)	NR, 9–18	NR	To ameliorate symptoms of mitral valve prolapse (e.g., chest pain, fatigue, etc.)	NR	8 sessions (40 minutes) + twice daily practice (15 minutes)	Small sample size, inadequate duration of treatment, lack of compliance in home practice
Self-management, progressive relaxation	Vazquez (1993), UK [38]	CCT	9 (treatment)	10.81 (NR), 8–13	70%	To treat bronchial asthma	NR	6 weekly one hour sessions	Small sample size, patient heterogeneity may confound relationship between intervention and outcome
Various MB therapies	Ding (2017), AUS [39]	Cross-Sectional Survey	381	NR, 0–18	52%	Various, aimed to determine 12 month prevalence/nature of alternative therapy use in pediatric patients	NR	NR	Observational study Minimal details of AEs

CCT—controlled clinical trial; NR—not reported; RCT—randomized controlled trial; SAE—single-arm experimental study.

**Table 2 children-08-00358-t002:** Summary of adverse events following mind-body practices in pediatrics by severity grad.

First Author (Year), Country	MB Practice	# of AE (s)	Age/Sex	# of Study Partici-Pants	AE Description	Timing of AE	Outcome of AE	Results/Conclusion by Authors
**Severity Grade 3**								
Bianchi (2004), Italy [33]	Yoga	1	14F	1	Fracture of distal tibia	While attempting to assume “lotus” yoga position	Resolved with standard leg immobilization, casting, and rehabilitation	Yoga can result in severe damage in adolescents due to age and open growth plates
LeBaron (1985), US [24]	Hypnosis	1	18M	1	Spontaneous intra-abdominal bleed	A few hours after administration of hypnotic scale	Resolved by hematologist treatment	Physiological effects of hypnosis in hemophilia population is unknown and potential risk may exist
Smith (1984), US [26]	Self-hypnosis	1	13F	1	Self-hypnosis misinterpreted as CNS deterioration in ALL case	Four days after learning self-hypnosis	Resolved with therapist’s help, returned to stable/alert state	Self-hypnosis needs a conscientious practice of the technique and appropriate communication with others
**Severity Grade 2**								
Armstrong (1976), US [36]	Biofeedback and prog-ressive muscle relaxation	1	17M	1	Depression and unavailability from therapeutic engagement	Post-intervention	NR	Removal of patient’s somatic complaint eliminated the only channel open to therapeutic engagement
Page (1990), US [25]	Hypnosis	1	18F	1	Apparent epileptic seizure	While in the hypnotic state	Resolved, normal EEG post event	Pre-induction precautions, omitting references to after effects, and careful observation during hypnosis suggested
St. Louis (2006), UK [29]	Transcendental Meditation	1	18F	1	Temporal lobe epilepsy (4 “spells” in a year and 3 generalized tonic-colonic seizures)	Following sleep deprivation and missed medication doses	Became seizure free for 6 months with medication and continued meditation practice	Further retrospective and prospective studies needed to determine whether meditation can precipitate epilepsy
**Severity Grade 1**								
Anbar (2005), UK [21]	Self-hypnosis	1	15M	1	Blue-tinted vision and concurrent penile erection	Half of the times therapy utilized	Continued to occur with self-hypnosis	Controlled studies with biological measurement of retinal blood flow after self-hypnosis may determine cause of blue-tinted vision
Bolek (2006), US [19]	Biofeedback	2	13F, 13M	16	Anxiety (*n* = 1) and foot pain (*n* = 1) due to weight issues on standing	During therapy	Anxiety improved with distraction by program’s video; discontinued therapy	Surface electromyography helps improve motor performance in treatment resistant children
Dahl (1988), US [20]	Biofeedback	2	NR/NR	3	Anxiety when aware of early seizure signals	During therapy	NR	Biofeedback reduced refractory seizure behaviour and paroxysmal EEG activity
Haber (1979), US [22]	Hypnosis	3	14M, 14M, 17M	8	Dissociated state (*n* = 1), depersonalization and anxiety (*n* = 1), increased anxiety (*n* = 1)	During therapy and post-hypnosis	Resolved with discontinuation and counseling	Hypnosis may have associated adverse events and did not appear to have any advantages over other therapeutic options
Huth (2004), Netherlands [28]	Imagery	2	NR/2M	36	Distress (*n* = 1), physical shaking (*n* = 1)	In anticipation of therapy; during therapy	Withdrew from the study	Imagery is associated with a reduction in post-operative pain and anxiety
Kellerman (1983), US [23]	Hypnosis	1	NR/1M	16	Feeling uncomfortable while practicing hypnosis	During therapy	Declined further treatment	Hypnosis has value in reducing procedural associated anxiety and discomfort in adolescent cancer patients
Page (1990), US [25]	Hypnosis	1	18M	1	Retroactive amnesia; unable to recall phone numbers	~100 minutes following hypnosis	Resolved by looking at numbers again, no further retroactive amnesia experienced	Suggest that therapists employ careful observation during their routine
Smith (1989), US [37]	Biofeedback, imagery, relaxation	1	NR/NR	20	Increased chest pain	Post-therapy	NR	Chest pain decreased at 6 months in mitral valve prolapse with biofeedback and relaxation/imagery treatment
Thygeson (2010), US [35]	Yoga	1	NR/NR	16	Dizziness	During yoga	Withdrew from study	Yoga is a feasible intervention and beneficial to adolescent patients and parents
Vazquez (1993), UK [38]	Progressive muscle relaxation	4	NR/NR	9	Increased drug consumption in emotionally-triggered asthma	During therapy	NR	Relaxation was found to be effective in emotionally-triggered asthma
Zarkowska (1989), UK [31]	Cue-controlled relaxation training	1	13F	1	Increased tic frequency from baseline	Post-intervention	Resolved with a trial of medication	Relaxation failed to reduce tic frequency
Zeltzer (1983), US [27]	Hypnosis	1	13M	9	Physical discomfort	During therapy	Discontinued therapy	The results support the efficacy of hypnosis as a means of reducing emesis
**Unclassified**								
Benavides (2009), UK [32]	Ashtanga yoga	4	NR/NR	14	Lower self-esteem (*n* = 2), Increased depression symptoms (*n* = 2)	Post-intervention	NR	Yoga may represent an alternative for weight loss and provide mental health benefits
Ding (2017), AUS [39]	Yoga (*n* = 2), massage (*n* = 1), hypno-therapy (*n* = 1)	4	NR/NR	381	Hypnotherapy: increased anxiety; NR for other AE	NR	NR	Alternative therapy use is common among pediatric ER patients. Parents who arrange alternative therapy have differing perceptions of its usefulness/safety from those who do not
McNally (2018), USA [30]	Relaxation	1	NR/NR	26	Worsened concussion symptoms	NR	NR	Brief cognitive behavioural intervention a promising treatment for children and adolescents experiencing persistent post-concussive symptoms
Moody (2017), USA [34]	Yoga	2	NR/NR	35	Avascular necrosis (*n* = 1) Acute splenic sequestration (*n* = 1)	NR	NR	Yoga is an acceptable, feasible and helpful intervention for hospitalized children with vaso-occlusive crisis

NR—not reported.

## Data Availability

All data was obtained from existing published material, which has been referenced.

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
