# Peer review of "Adverse Events of Mind-Body Interventions in Children: A Systematic Review"

_children, 2021, doi:10.3390/children8050358_

Round 1

Reviewer 1 Report

The authors present a useful, systematic review of adverse events associated with the use of mind-body interventions in children. This study's strengths are that the authors have performed a comprehensive search, according to the PRISMA guidelines, and that they have used CTCAE criteria for rating severity of AEs. The authors found 377 articles without safety assessment of not mentioning AE’s, 43 articles stating no AE’s had occurred, and 21 articles reporting a total of 37 adverse events.  The authors then focus on discussing these reported events, all in Grade 1 to 3. Since 377 articles did not report AE’s, I agree with the authors that future studies should focus more on reporting adverse events, which will make statements about the safety of mind-body interventions more solid. 

Nevertheless, the article would benefit from discussing the fact that only 37 AE’s were mentioned in a group of 2019 children/ adolescents (1405 + 208 + 406). Most AE’s were very mild, no grade 4 or 5 AE’s were reported with only three grade 3 and three grade 2 adverse events, suggesting that mind-body interventions are relatively safe therapies compared to, for example, pharmacological treatment. In my opinion, that deserves some attention in the discussion.

I am also missing the author’s comments on the causality. Several of the adverse events may also have been caused by the underlying disease.

One last remark: the databases were searched from inception to August 2018. It is now already 2,5 years later and I think that the authors should try to expand their search to March 2021. Generally spoken, studies are nowadays of higher quality and reporting harms is part of the CONSORT guidelines for RCT’s, which may increase the chance of finding more studies reporting the occurrence of AE’s.

Reviewer 2 Report

This article serves as a good reminder that in order to really appreciate the benefits and risks of mind body therapies, research and documentation must address both positive and negative outcomes. I think it is a reach; however, to claim that they are the first study to report the safety of these interventions. The study shows that safety data is under-reported, not if specific MBT's are safe or not. 

I have a few questions for the study authors.

  1. How did you select your endpoint for data selection? Was this work originally done in 2018? If this is the case, given that over 2 years have passed since that date and this is a systematic review, I would suggest updating your literature review so you can present the best and most up to date data if possible.
  2. Include a description of the primary MB therapies you include - some readers may not know what biofeedback / clinical hypnosis entails. This report groups them in the same bucket as "MBT" but they are actually all quite different and as such would pose different degrees of risk. I would also add some information as to what conditions pediatric patients are most likely to use MB therapies for and why in your introduction.
  3. In table 1, I would suggest grouping the study by MB intervention rather than author last name. Then it would be easier to compare AE reported for a particular modality and quantify how many studies of that modality are in your review.
  4. In table 2, include a column with the N of the study (or change the number column to 1/N) to help put number of participants effected into better context
  5. For the first sentence of your discussion and around line 258, it is an overstatement to say you are the first study to examine the safety of these interventions and report all adverse events. This is the first study to demonstrate that AE are incompletely / under-reported in pediatric MB studies and summarizes what AE have been reported to date.
